# Effects of Facial Muscles Exercise on Mental Health: A Systematic Review

**DOI:** 10.3390/ijerph182212216

**Published:** 2021-11-21

**Authors:** Rumiko Okamoto, Toshie Manabe, Katsuyoshi Mizukami

**Affiliations:** 1Faculty of Health and Sport Sciences, University of Tsukuba, Tokyo 112-0012, Japan; mizukami.katsuyoshi.ga@u.tsukuba.ac.jp; 2Nagoya City University Graduate School of Medical Sciences, Nagoya 467-8601, Aichi, Japan; manabe@kklabo.gr.jp

**Keywords:** facial expression, facial muscles, self-care, mental health, depression

## Abstract

Mental disorders are increasing worldwide. Previous research has reported an association between mental health and facial expressions. Face-to-face communication, specifically, is majorly affected when wearing face masks for a long time because of the COVID-19 pandemic. However, there have been no systematic reviews of facial muscles exercise intervention studies for mental health. Thus, evidence of their effect on mental health is unclear. This review aimed to evaluate the current evidence of the effectiveness of voluntary facial muscles exercise to improve some parameters of mental health. We implemented a systematic review of experimental studies (published between 2007 and 2018, 10 years before we decided to start this review). Of the 61,096 references screened, seven studies reported that facial muscles exercise may help to improve some parameters of mental health. Moreover, the study quality was assessed, and we extracted sub outcomes for mental health. Non-coherent results of seven experimental studies were included in this review. Voluntary facial muscles exercise may help improve depressive symptoms, mood, and reduce the level of chronic stress. However, due to the low quality of analyzed studies, further studies are needed to confirm the mental health benefits of a facial muscles exercise program.

## 1. Introduction

The COVID-19 outbreak is stressful for people and communities in the world. Specifically, face-to-face communication is majorly affected by wearing face masks for a long time because of the COVID-19 pandemic. A high prevalence of psychological morbidities has been reported among individuals who are directly or vicariously exposed to life-threatening situations [1,2,3].

Mental disorders are increasing worldwide not only due to the COVID-19 but also to other causes. Between 1990 and 2013, the number of people with depression and/or anxiety increased by nearly 50%, from 416 million to 615 million. In 2016, close to 10% of the world’s population was affected, and mental disorders accounted for 30% of the global, nonfatal disease burden [4]. Recently, there has been increased interest in noninvasive and nonpharmacological therapies [5]. There is a minimum amount of physical activity for health benefits, and these benefits increase with the amount of exercise; however, beyond a certain level, the adverse effects outweigh the benefits [6]. Therefore, this study focused on facial exercise not physical exercise.

The facial feedback hypothesis [7,8] states that facial movement may influence emotional experience [9]. Facial expressions for basic emotions (e.g., surprise, happiness, anger, fear, sadness, disgust) have been reported to be common facial expressions in the world [10]. Certain facial movements can be controlled voluntarily, while others occur primarily during “genuine” emotions. For example, voluntary smiles (e.g., for business purposes, without any particular emotional involvement) usually consist only of the upward curving of the lips, whereas spontaneous smiles due to positive emotions also include the eyes (the so-called “Duchenne smile”), characterized by a rising of the cheeks and the appearance of crow’s feet next to the eyes [11,12]. These two facial expressions are mediated by different neural pathways. Voluntary smiles are initiated in the motor cortex and through the pyramidal motor system, whereas involuntary smiles arise mainly from the subcortical nuclei and through the extrapyramidal motor system. An unexpected byproduct of this research has been the observation that voluntarily producing and holding an expression can induce the corresponding emotion [13,14]. This effect is more pronounced when a person pays specific attention to activate voluntary muscles that are usually only used involuntarily (e.g., the Duchenne smile) [15,16]. On the other hand, a study reported that the botulinum toxin of the corrugator muscle resulted in decreased activation of the brain regions implicated in the emotional processing and emotional experience, namely, the amygdala and the brainstem [17].

Facial movements are unique and present considerable differences compared with the rest of the human motor repertoire, in particular with movements of the upper limbs [18]. The central control of facial movements is complex and relies on multiple parallel systems (for example, the voluntary and the affective systems), which are segregated anatomically and functionally up to a very distal level, such as up to the facial nucleus. Previous research suggests aerobic exercise-induced increases in the skeletal muscle PGC-1α1 may be an important regulator of KAT expression in the skeletal muscle, which, via modulation in plasma kynurenine levels, may alleviate stress-induced depression and promote hippocampal neuronal plasticity [19]. On the other hand, the neurons contributing to the control of facial muscle movements are distributed over several brainstem regions [20]. The facial muscles exercise is controlled by voluntary and involuntary mechanisms. The autonomic nervous system (ANS) is composed of the sympathetic nervous system (SNS) and the parasympathetic nervous system (PNS). The main peripheral pathways of the PNS are within the vagus nerves. The myelinated vagus, one of the vagus nerves, found only in mammals, supports social engagement and engenders feelings of safety. Myelinated vagal efferent fibers from the nucleus ambiguus serve as the vagal brake, which enables rapid control of the heart rate (HR) by increasing the vagal tone to reduce the HR and blood pressure or by decreasing the vagal tone to accelerate the heart rate [21]. Voluntarily controlled facial movement patterns may affect the ANS and HRV, such as breathing. In addition, facial muscles exercise activates the motor component of the ventral vagal complex, which originates in the nucleus ambiguus and regulates and coordinates the muscles of the face and head with the bronchi and heart. These connections help orient the person towards human connection and engagement in prosocial interactions and provide more flexible and adaptive responses to environmental challenges, including social interactions [22]. The social communication system involves the myelinated vagus, which serves to foster calm behavioral states by inhibiting sympathetic influences on the heart and dampening the hypothalamic–pituitary–adrenal (HPA) axis [23]. Facial expressions are also part of stereotyped physiological responses to particular affective states, involving both the ANS and the somatic systems, which are controlled by the so-called emotional motor system [24,25]. Thus, in view of the close association among respiration, ANS activity, and emotions by facial muscles exercise, it can be thought that an individual has the ability to change emotional states using spontaneous control of breathing and thinking. Emilou et al. (2017) [26] indicated that mental health mediated the association between each psychosocial resource and a younger appearance. A recent study suggested that functional electrical stimulation of the facial muscles responsible for a ‘genuine smile’ rather than depressing the activity of the corrugator muscle and that it improved symptoms in individuals with a major depressive disorder [27].

The aim of this systematic review was to evaluate the current evidence of the effectiveness of voluntary facial muscles exercise on improving mental health.

## 2. Materials and Methods

This systematic review was conducted according to the Preferred Reporting Items for Systematic Reviews and Meta-Analysis (PRISMA) statement [28]. A predefined protocol was not registered. Institutional review board approval and patient consent were not required because of the review nature of this study.

### 2.1. Search Strategy

Two investigators (RO and TM) independently searched for eligible studies in PubMed and Ichu-shi Web, which is a Japanese journal database [29] published between database inception and June 2019. We used the following key words: “(face OR facial OR “facial muscle*” OR “facial mimic*” OR “facial move*” OR “facial expression” OR “facial appear*”) AND (exercise* OR training OR fitness OR therapy OR treatment) AND (“mental health” OR depres* OR mood OR anxiety* OR stress* OR “negative behavior” OR “emotional exhaustion” OR well-being OR QOL OR engagement OR burnout OR smil* OR laugh*)”. The search was limited to studies written in English. We also reviewed the reference lists of eligible studies using Google Scholar and performed a manual search to ensure that all appropriate studies were included.

### 2.2. Eligibility Criteria and Outcome Measures

Studies fulfilling the following selection criteria were included in the systematic review: (1) study design and language: randomized controlled trials (RCT), quasi-experimental studies, and before and after studies in English language publications; (2) population: people with or without clinical diagnoses of a physical or mental illness; (3) primary outcome variables: the effectiveness of facial exercises for mental health.

Studies were excluded based on the following criteria: (1) studies that only had abstracts; (2) studies where the outcome variable was not reported; (3) studies that tested smiling and laughing.

### 2.3. Data Extraction

Two reviewers extracted the data independently. After removing redundant articles, titles, and abstracts, full-text articles were investigated. We extracted the following data: study design, study period, study site, study setting (school, clinic, hospital, office, or community-dwelling facility), inclusion/exclusion criteria of each study, information source, type of facial exercises (exercise or training with massage or breathing), and general background. The outcome variables were extracted into predesigned data collection forms. We verified data accuracy by comparing the collection forms of each investigator; any discrepancies were resolved through discussion together with the other author (KM).

### 2.4. Data Analysis/Synthesis

Due to the high degree of heterogeneity relating to the study outcome measures, study populations, and intervention length/intensity, a meta-analysis was not performed.

## 3. Results

### 3.1. Study Selection and Characteristics

A PRISMA flow diagram for the search process and results (Figure 1).

Of the 61,096 references screened, seven studies reported the effectiveness of facial exercises for mental health. These 61,096 articles were screened after excluding the duplicates. After reading the titles and abstracts, 61,057 articles were excluded. Then, 20 articles were excluded for lack of facial exercise, and a further 10 articles were excluded after reading the full texts for eligibility. Seven studies (Nomura et al., 2007 [30]; Uchida and Arai, 2010 [31]; Konecny et al., 2011 [32]; Chang et al., 2013 [33]; Chuchuen et al., 2015 [34]; Hyoung and Sung, 2016 [35]; and Okamoto and Mizukami, 2018 [36]) met the criteria for inclusion. The authors used the procedure ver.4 for creating the medical practice guideline provided by the medical information service “Minds” of the Japan Council for Quality Health Care as a reference for the classification of the evidence level [37] (Table 1).

Two of them were before–after studies [30,31], one was a quasi-experimental study with comparison groups [35], and four were RCTs [32,33,34,36]. The studies were conducted in five countries: Czech Republic [32], Taiwan [33], Thailand [34], Korea [35], and Japan [30,31,36]. The studies involved a variety of participants, including grade seven students of a public junior high school [33], private office workers [34], nursing care staff [30], community-dwelling older adults [36], Parkinson’s disease patients [31], and facial palsy patients [32,35] (Table 2).

### 3.2. Interventions

The term of the facial muscles exercise sessions varied from 2 weeks to 3 months. The duration of the sessions also varied; two studies were less than or equal to 5 min [30,31], one study was 20 min [35], one study was 30 min [36], one study was 45 min [33], one study was 1 h [34], and the RCT study did not report the duration of each session [32]. Furthermore, we calculated the total amount of time (in minutes) spent on the training program in each study. The shortest was 120 min [35], and the longest was 1260 min [31]. The three studies [30,32,36] took about the same 720–750 min. We reported that the intensity of the facial muscles exercise is 2.5–3 METS (e.g., yoga, stretching) by an expiratory gas analysis apparatus (FIT2000, Nihon Kohden, Tokyo, Japan) [36].

### 3.3. Execise Program

The program of facial muscles exercise varied in the seven studies. Two studies were smile muscles exercises [30,34], one study was facial tai chi [33], and four studies were facial muscles exercises and included other training [31,32,35,36].

### 3.4. Outcome Measurements

All studies measured preintervention and postintervention results. In addition, all of the studies used different scales or measurements for mental health. Nomura et al. (2007) [30] evaluated nursing care staff by the Brief Job Stress Questionnaire (BJSQ; Ministry of Health, Labor and Welfare, Japan, 2015) [38]; Uchida and Arai (2010) [31] measured depression with the Self-Rating Depression Scale (SDS; Zung, 1965 [39]); and the study of Konecny et al. (2011) [32] used the Beck Depression Inventory-Second Edition (BDI-II; Beck et al., 1996 [40]) and a self-report questionnaire to assess depression symptoms. In terms of enhancing positive mood, in the study of Chang et al. (2013) [33], the Chinese Humor Scale (CHS) scale (Hsieh et al., 2005 [41]) was used to assess general positive mood (Chang et al., 2013 [33]); Chuchuen et al. (2015) [34] assessed the stress level of private office workers using the Suanprung Stress Test-60 (SPST-60; Mahatnirunkul et al., 1997 [42]); Hyoung et al. (2016) used the Center for Epidemiological Studies Depression Scale (CES-D; Radloff, 1977 [43]) for facial palsy patients; and we [36] used the General Health Questionnaire-12 (GHQ12; Goldberg, 1997 [44]) in community-dwelling older adults.

### 3.5. Risk of Bias and Quality Assessment

Overall, the evidence generated by the studies involved was weak. All studies had relatively small sample sizes, and the risk of selection bias was unclear but possible. And all of the studies included were also affected by a variety of other potential biases, as they provided little information about the in-depth content of the intervention, participant compliance, or adverse events.

### 3.6. Effect of Facial Muscles Exercise on Mental Health

#### 3.6.1. Depressive Symptoms

In Uchida and Arai’s study (2010) [31], the SDS scores significantly decreased between the pretest (50.2) and the posttest (41.6) (*p* = 0.00003). In the study of Konecny et al. (2011) [32], significant improvements in the intervention group (IG) could also be seen in the evaluation of depression according to the BDI-II. There was a significant difference in the mean values of the BDI-II before and after rehabilitation in the IG (14.3 [SD 5.1]) and in the CG (6.9 [SD 5.1]). The CES-D scores were significant in the IG (*p* < 0.001). We showed that the change in the GHQ-12 scores was significant in the IG (*p* = 0.003) [36].

#### 3.6.2. Mood

Chang et al. (2013) [33] assessed mood states using the Face Scale after each session, and significant differences were observed in the IG during sessions 2, 3, 4, 6, and 7 (*p* < 0.00).

#### 3.6.3. Stress

Two studies [30,34] reported the effects of facial exercise on stress. Nomura et al. (2007) [30] reported that job overload (one of subscales measured using the BJSQ) significantly decreased after facial exercise in the intervention group. In the study of Chuchuen et al. (2015) [34], there were no significant differences in sensitivities to stress, sensitivities to arousal events, and signs and symptoms of stress. Above all, it was observed that sensitivities to arousal events decreased from extremely high (61.10 + 16.22) to medium (49.26 + 20.01); however, this was not significant.

#### 3.6.4. Humor

In the study of Chang et al. (2013) [33], the experimental and control groups were compared using humor (CHS) after completing facial exercise. The mean CHS scores significantly increased in the IG from 78.15 (± 19.13) to 85.27 (± 20.00) after facial muscles exercise (*p* = 0.004), and one of the subscales of the CHS, Humor Creativity (HC), also showed a significant increase for the IG from 39.59 (± 11.46) to 44.38 (± 12.30) after facial exercise (*p* = 0.003). The ANCOVA results showed a significant influence of facial exercise on CHS (*p* = 0.01) and HC (*p* = 0.01). There were no significant changes in the CG for any of the psychological measures.

#### 3.6.5. Immunological and Physiological Effects

In the study of Chang et al. [33], cortisol levels were significantly different pretest 48.35 (± 12.53) to posttest 38.50 (± 13.10) (*p* = 0.001) in the IG. The ANCOVA results showed no significant changes (*p* = 0.058).

#### 3.6.6. Self-Reported

The study of Nomura et al. (2007) reported responses such as “I relaxed the stiffness of the face after facial exercise”, “I wanted to do facial exercise a little longer”; however, on the other hand, negative responses, such as “I can’t feel the effects of facial exercise”, were also reported [30]. In our study (2018), 95% of participants responded with “I would like to continue this facial exercise” and 72% of participants answered “My heart felt lighter” and “I realized the effects of facial exercise” [36].

## 4. Discussion

### 4.1. Findings

This systematic review was performed to compare the effectiveness of the facial muscles exercise on mental health. There was a significant impact on the SDS, BDI-Ⅱ, CHS, CES-D, GHQ-12, and Face Scale. However, with the BJSQ, Self-Restricted Type Behavior Traits Scale, Emotional Support Network Scale, RSE, SPST-60, and PGC not statistically significant changes were noted. As a result, we extracted the sub-outcomes of the improvements to depression, mood, stress, and humor, and a decrease in cortisol levels. Improvement by the scales (SDS, BDI-Ⅱ, CES-D, GHQ-12) in the research extracted in this systematic review suggested facial muscles exercise have an effect on depressive symptoms. These findings of voluntary movements are similar to the previous study that showed passive movements have an effect on mental health [26].

Future facial muscles exercise studies should include more participants and consider measuring mental health as a major consequence. In addition, it is important to coordinate the duration and intensity of the intervention with the same program as in previous studies to continue the study. To minimize the effects of performance bias, future studies should also consider the use of objective physiological measurements related to mental health. Studies should also use a follow-up period of up to six months to confirm the sustainability of the benefits to mental health. Future studies should consider incorporating a monitoring system and using participant interviews to investigate the participant’s experience, as the review did not report information on adverse effects. It also requires clear information about the specific content of the facial muscles exercise group (i.e., using a manual approach or detailed intervention protocol) so that other researchers can faithfully reproduce the intervention in future studies. In addition, this information should include a detailed description of each session content, optimal group size, number of sessions, session frequency, length of time for each session, and possible flexibility or steps in facilitating the group.

### 4.2. Limitations of the Review

Our review had some limitations. First, studies without standardized results measurements and papers published outside of English were excluded. Second, we adopted strict criteria for “facial muscles movement” to increase the comparability of the study. As a result, we excluded studies that described the interventions such as “laughter yoga” or “laughter therapy”, but some of these interventions may have resembled facial muscles movements. In addition, a meta-analysis could not be performed due to the small number of studies and the large heterogeneity associated with the study population, facial muscles exercise program, and measured results. In addition, this conclusion should be interpreted with caution, as the systematic review included a number of low-quality studies (without a small N or the exact same exercise program) and studies at high risk of bias.

Therefore, although a summary of these studies suggests the facial muscles exercise improve some parameters of mental health, the methodological shortcomings of these studies make it hard to assert the effects on mental health. As some of the included studies in this systematic review used nonrandomized study designs, had very small sample sizes, or were otherwise at high risk of bias, many of these results could be spurious findings or other statistical artefacts, such as regression to the mean or nonspecific factors. For the future, it is important to continue study of high quality.

## 5. Conclusions

As far as we know, this is the first review of the effects of facial muscles exercise on mental health. The results of this systematic review suggested that voluntary facial muscles exercise may help improve depressive symptoms, mood, and reduce level of chronic stress. However, due to the lack of available evidence, it was difficult to determine whether facial muscles exercise was effected on mental health. Further studies could implement to evaluate the effects of other mental health (cognitive function, anxiety symptoms, burn out syndrome, etc.). It is also necessary to consider positive mental health effects such as self-esteem and well-being. Facial muscles exercise showed potential as a treatment, however, currently there is insufficient evidence to support its effectiveness in improving mental health compared with other group-based facial movement interventions. In the future, continued research, which is rigorously designed, and pile up evidence of high quality are required.

## Figures and Tables

**Figure 1 ijerph-18-12216-f001:**
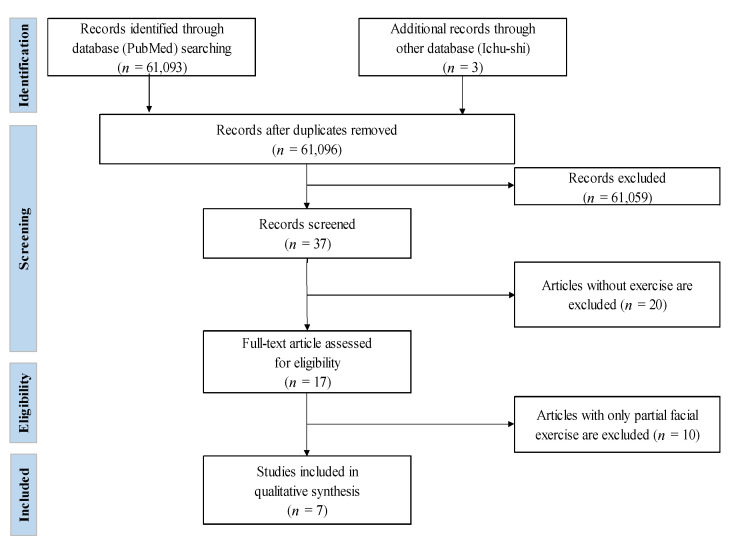
PRISMA flow diagram.

**Table 1 ijerph-18-12216-t001:** Classification standard of the evidence level.

Level of Evidence	Intervention
Ⅰ	Meta-Analyses of Randomized Controlled Trials/Systematic Reviews
Ⅱ	Randomized Controlled Trials
Ⅲ	Controlled Clinical Trials
Ⅳa	Cohort Studies
Ⅳb	Case-Control Studies
Ⅴ	Case series/Case reports
Ⅵ	Expert Opinions/Non-EBM Guidelines

**Table 2 ijerph-18-12216-t002:** Characteristics of included studies.

Authors (Year)	Country	Participant Characteristics	Intervention (Total Time)	Exercise Program	Study Design	Measuring Method	Outcome Measures	Results	Evidence Level
Nomura et al. (2007) [30]	Japan	*n* = 33 (Nursing care staff), Average age: 37.4 ± 5.33 years	5 min/day, 5 days/week for a month (before morning meeting) (750 min)	Smile muscles exercise	Before–after	Psychological measures	(1) The Brief Job Stress Questionnaire; (2) Self- Restricted Type Behavior Traits Scale; (3) Emotional Support Network Scale	(1) The workload (one of subscales) score improved in the IG (*p* < 0.01); (2) Not statistically significant; (3) Not statistically significant	Ⅳb
Uchida and Arai (2010) [31]	Japan	*n* = 31 (females: 18, males: 13, Outpatients with Parkinson’s disease), Average age: 67.0 years, The Hoehn and Yahr Scale: Stage Ⅱ–Ⅴ, ICD10: F32	a few min/day, every day for 2 months (at home) (1260 min)	(1) Facial muscles training; (2) Positive thinking training	Before–after	(1) Analysis of facial emotion; (2) Psychological measures	(1) FACS; (2) The Unified Parkinson’s Disease Rating Scale (UPDRS); (3) Self-Rating Depression Scale (SDS)	(1) The elevation of corners of mouth were significantly improved in the IG (*p* = 0.00003); (2) The UPDRS scores showed significantly improved between pretest 5.58 and posttest 1.55 (*p* = 0.00000); (3) The SDS scores showed significantly decreased between pretest 50.2 and posttest 41.6 (*p* = 0.00003)	Ⅳb
Konecny et al. (2011) [32]	Czech Republic	*n* = 99 (females: 46, males: 53, Facial palsy patients), Average age: 61.8 years	20 min/day, for a month (720 min)	(1) Speech therapy; (2) Rehabilitation exercises; (3) Facial exercise therapy include breathing rehabilitation; (4) Speech therapy; (5) Rehabilitation exercises	RCT	(1) Analysis of facial movement; (2) Analysis of facial emotion; (3) Psychological measures	(1) House–Brackmann Grading Scale (HBGS); (2) 2D video analysis; (3) BDI-II	(1) A significant difference in the mean values on the HBGS before and after rehabilitation of 1.66 (SD 0.55) was observed in the experimental group and of 0.59 (SD 0.57) in the control group. (2) There were significant difference in the changes in distances between the corner of the mouth and the earlobe in the experimental group (11.5 mm (SD 3.50)) and in the control group (2.0 mm (SD 2.30)). (3) Statistically significant improvements in the experimental group can also be seen in the evaluation of depression according to the BDI-II. There was a significant difference in the mean values of the BDI-II before and after rehabilitation in the experimental group (14.3 (SD 5.1)) and in the control group (6.9 (SD 5.1)).	Ⅱ
Chang et al. (2013) [33]	Taiwan	*n* = 67 (females: 33, males: 34, students of a public junior high school, 7th grade)	45 min/day, 8 sessions for 2 months (360 min)	(1) Facial tai chi; (2) Nothing	RCT	(1) Psychological measures; (2) Immunological measures	(1) Rosenberg’s Self-Esteem Scale (RSE); (2) The Chinese Humor Scale (CHS) (3) The Face Scale (FS); (4) ELISA; (5) Blood pressure, Heart rate variability, LF/HF	(1) Not statistically significant; (2) The CHS and Humor Creativity (one of subscales) scores improved in the IG (*p* = 0.004; *p* = 0.003); (3) Not statistically significant; (4) Cortisol showed significant differences between pretest 48.35 (SD 12.53) and posttest 38.50 (SD 13.10) in the IG (*p* = 0.001). (5) Not statistically significant	Ⅱ
Chuchun et al. (2015) [34]	Thailand	*n* = 38 (private office workers), Age: 25–60 years	60 min/day, 3 days/week, 8 sessions for 2 months (480 min)	(1) Smile muscles training; (2) Nothing	RCT	Psychological measures	The Suanprung Stress Test-60 (SPST-60)	(1) Not statistically significant; (2) No significant difference was found in the mean scores of the level of stress between the CG and IG. However, the sensitivities to the arousal events in the IG had a tendency to decrease	Ⅱ
Hyoung and Sang (2016) [35]	Korea	*n* = 70 (females: 45, males: 25, Outpatients with Facial palsy), Age: < 40: n = 16, 40–49 years: n = 15, 50–59 years: n = 19, > 60: n = 2	20 min/day, 3 days/week, 2 weeks (120 min)	(1) Facial muscles exercises; (2) Facial massage (3) Nothing	Quasi-experimental	(1) Analysis of facial movement; (2) Psychological measures; (3) Reading measures	(1) House-Brackmann Grading Scale (HBGS); (2) Facial Nerve Grading Scale (FNGS); (3) Reading Time; (4) Palsy subjective symptoms; (5) CES-D	(1) Not statistically significant; (2) The FNGS scores were statistically significant in the IG (*p* < 0.001); (3) Not statistically significant. (4) The CES-D scores were statistically significant in the IG (*p* < 0.001)	Ⅲ
Okamoto and Mizukami (2018) [36]	Japan	*n* = 53 (females: 51, males: 2, Community-dwelling older adults), Average age 78.3 ± 5.3 years	30 min/day, 2 days/week, 24 sessions for 3 months (720 min)	(1) Facial acupressure; (2) Facial muscles exercises; (3) Facial yoga; (4) Facial massage; (5) Nothing	RCT	(1) Psychological measures; (2) Tongue pressure; (3) Analysis of facial emotion	(1) GHQ-12; (2) PGC Moral Scale; (3) Tongue pressure (4) Face Reader™	(1) The GHQ-12 scores were statistically significant in the IG (*p* = 0.003). There was a simple main effect; (2) Not statistically significant; (3) The tongue pressure scores were statistically significant in the IG (*p* = 0.036); (4) The happiness (smile) scores were significantly improved in the IG (*p* = 0.003)	Ⅱ

Mental health: GHQ-12; Depression: SDS, BDI-II, CES-D; Stress: BJSQ, SPST-60; Humor: CHS; Subjective Well-Being Scale: PGC Moral Scale; Mood: Face Scale, Face Reader™; Self-Esteem: RSE.

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
