# Peer review of "Effects of Facial Muscles Exercise on Mental Health: A Systematic Review"

_ijerph, 2021, doi:10.3390/ijerph182212216_

Round 1

Reviewer 1 Report

Congratulations on the work put in the paper. However, still there is a huge problem with inadequate conclusions drawn from the study.

As You wrote in response to me:

„There were a significant impact on SDS, BDI-â…¡, CHS, CES-D, GHQ-12 and the Face Scale. But, BJSQ, Self-restricted Type Behavior Traits Scale, Emo-tional Support Network Scale, RSE, SPST-60, PGC were not statistically significant changes were noted. As a result, we extracted sub-outcomes of improvements to depression, mood, stress, and humor, and a decrease in cortisol 3 levels. improvement by the scales(SDS, BDI-â…¡, CES-D, GHQ-12) in the research extracted in this systematic review suggested facial muscles exercise effect on depressive symptoms.”.

If Your conclusion is that facial muscles exercise might presumably reduce depressive symptoms severity, then why You have chose to write in Your conclusions in the current form of the paper :” The effectiveness of facial muscles exercise were depressive symptoms, mood, stress, humor, and a decrease of cortisol levels. The other hand, the overall level of evidence was weak due to poor study quality and risks of bias. Facial muscles exercise shows potential as a treatment, however, currently there is insufficient evidence to support its effectiveness in improving mental health compared with other group-based facial movement interventions. In the future, continue research, rigorously designed and pile up evidences of high-quality are required.” In the abstract and “The results of this systematic review suggest that voluntary facial muscles exercise may help improve mental health. However, it is not possible to draw firm conclusions. Also, due to the lack of available evidence, it was difficult to determine whether facial muscles exercise was effected for mental health.” In the conclusions in the main text?

First, conclusions in the main text seem to be contradictory.

Second, please be specific as much as possible in Your conclusions. If Your conclusions are that facial exercise program might presumably lead to reduce depression symptoms severity, please exactly phrase it. If further research are needed to examine facial muscle training program with sham intervention/other types of interventions, please phrase it as a conclusion. The same if You conclusion is that facial exercise program might presumably lead to reduce depression symptoms severity and level of chronic stress, please exactly phrase it and add what further studies could implement to validate this result etc.

Author Response

Thank you so much for your review. Please find attached about author's reply.

Reviewer 2 Report

This is a systematic review of the biomedical literature on the effectiveness of an intervention (facial muscles exercise) on mental health parameters. The work followed the PRISMA format for this type of study. The text is presented coherently, following the scientific methodology. However, the adjustments below are necessary.

  • In the Abstract, the conclusion (line 22) would need to be written with much more caution and humility; e.g., replace 'treatment' with 'health intervention,' among other phrasal attenuation.
  • Explain why the period between 2007 and 2018 was chosen as the publication date of the papers to be examined.
  • In the Introduction: between the extensive physiological explanation and the objective of the study, there should be one or more paragraphs about such intervention: say who structured it, whether and where it has been applied, etc.
  • Subsection 3.6.1 talks about depressive symptoms, but some paragraphs talk about stress. I imagine stress could have a specific sub-section.
  • Subsections 3.6.2 and 3.6.3 are missing; I understand that all 3.6 subsections should be renumbered.
  • The passages with 'improve mental health' could be replaced by 'improve some parameters of mental health' (e.g., line 293)
  • Specific corrections:
    - Line 18: fragmented sentence
    - Line 110: some words in strikethrough
    - Line 174: some highlighted words
    - Abbreviations not written in full: PGC, KAT, HRV, BDI, among many others
    - Table 2 has a cut-off part of the PDF file; unlike a Word file, the information was lost.

Author Response

(The authors gave the same response as above.)

Round 2

Reviewer 1 Report

Congratulations on changing conclusions in the main text.

I would also add a general information on the quality of analysed studies as “ Due to the low quality of analysed studies, further studies are needed to confirm mental health benefits of facial muscles exercise program”, if You believe that the statement is right. I would strongly suggest to change conclusions from abstract also. I would suggest to put conclusions from the main text to conclusions in the abstract, what about those two sentences?

„Voluntary facial muscles exercise may help improve depressive symptoms, mood and reduce level of chronic stress. However, due to the low quality of analysed studies, further studies are needed to confirm mental health benefits of facial muscles exercise program”

In my humble opinion, after fixing conclusions from abstract and  extensive editing of English language the paper would be ready to publish. The latter is important, because some paragraphs are hard to understand, however I believe that it would be fixed by paper editing.

Author Response

Thank you so much for the review, please find attached about author's reply.

Reviewer 2 Report

Overall, the authors were sensitive to the opportunities for improvement cited by me.
One point that was not met was talking more about such intervention: who structured it, whether and where it has been applied, etc. 
The authors' response was: "the developers and programs of the facial muscles program were inconsistent and couldn’t be written."
I did not understand well this objection. Maybe it has to do with developers' copyright (???).
It would be important data to convince readers that this intervention did not come "out of the blue".
However, the lack of this information does not invalidate the research and its results.
In this sense, the paper fulfill quality requirements to be published.

Author Response

Thank you so much for the review, please find attached about author's reply.

This manuscript is a resubmission of an earlier submission. The following is a list of the peer review reports and author responses from that submission.

Round 1

Reviewer 1 Report

The current version of manuscript needs to be improved in term of English language and style. The current form still did not resolve all previous flaws:

From previous review:

Major issues:

  1. „Pre-67 vious research suggests aerobic exercise-induced increases in skeletal muscle PGC-1α1 68 may be an important regulator of KAT expression in skeletal muscle, which, via modula-69 tion in plasma kynurenine levels, may alleviate stress-induced depression and promote 70 hippocampal neuronal plasticity(Ruegsegger et al., 2018) [19]. In addition, facial muscles 71 exercise activates the motor component of the ventral vagal complex, which originates in 72 the nucleus ambiguus, regulates and coordinates the muscles of the face and head with 73 the bronchi and heart. These connections help orient the person towards human connec-74 tion and engagement in prosocial interactions and provide more flexible and adaptive 75 responses to environmental challenges including social interactions (Sullivan et al., 2018 76 [20]).”

Please be more precise here. Is Your hypothesis is that some of the facial muscle exercise effects on some biochemical pathways are similar as in more traditional approach to aerobic training engaging large muscle groups as walking/running on treadmill cycling etc? What about the difference of amount of muscle groups engaged? Are the more complex movements more effective in most of the downstream biochemical effects? Is facial muscle exercise similar to continuous form of aerobic exercise or more to the interval training? What about those effects on autonomic nervous system? Could You be more specific? I suppose that the general consensus in the physiology of exercise is that during first seconds of aerobic exercise there is inhibition of vagus heart input, therefore heart rate could increase. With the further increase of exercise intensity there is increasingly stronger activation if sympathetic nervous system. So how facial muscle exercise activate ventral vagal complex? Could You explain that? Are You describing acute exercise effects session or long-term effects of exercise programme? Is the heart rate lowers during facial muscle exercise? So why there is an increase in METS as You state?

I suppose that facial muscle exercise might be a really interesting approach. Bot if You are referring to polyvagal theory as the mechanism underlying its effects, You should describe it more. It would be the best if You would be able to do so in comparison to more common forms of physical exercise. What about groups of patients that are not able to undergo “traditional” aerobic exercise training? Patients with CFS with sever post-exertional malaise, patients with movement disorders? Maybe they can be engaged is such training programme involving face muscles only?

  1. „Two investigators (RO and TM) independently searched for eligible studies in Pub-91 Med, Ichu-shi Web, which is a Japanese journal database[24] of Systematic Reviews”

It is nice that You have used more than one database. However, one can have an impression that You have not clearly phrased the aim of the search. Why did You have included database for systematic reviews? Were You looking for studies intervention bases studies? It would be the best to do systematic review on RCTs, however I suppose that there is lack of high quality studies in this area, what should be mentioned in the limitations. But why searching for systematic reviews? Is the current paper an umbrella review? Maybe it would be better for before-after intervention based studies? Or intervention based studies with control groups? What was the aim of the search? If not looking for systematic reviews, that why using database for systematic reviews?

There are two table 1 in the current version of paper. Why put “Table 1. Classification standard of evidence level”?

  1. Considering limitations of the analysed trials, Your conclusions “The outcomes of this systematic review suggest that voluntary 269 facial muscle exercises may help to improve mental health” are too strong. What mental health indicators could be improved? Are those conclusions specific to psychiatric disorders? You should be more specific here, conclusions should be firmly based on You analysis. Please be as specific as You can.

Minor issues:

“reported that the intencity of facial muscles exercise is 3-3.5 METS( similar to brisk walk-157 ing) [31].”

Once again, this section needs to be improved. The idea is to show the reader the high variability in training programs specifics included in studied that You have analysed. For instance, there were some that lasted for two weeks. In literature on hypertrophic training 2 weeks would be too ow amount of time to show any significant improvement in muscle mass. On the other hand some training programme(s) lasted for 3 months. Maybe the best would be to calculate total amount of time (in minutes or hours) spent on training programme in each study? So if it was 2 weeks and 2 session for 60 minutes per week it would be 4 session in total for 60 minutes, what would give 4*60=240 minutes of total amount of time spend on exercise. Could You report that measure for each analysed study? Are You noting some relationship between time spend on training programmes and effects? Or maybe even 2 weeks could be effective in improving primary outcomes? What about the intensity of exercise? Is that was reported? If not, it is a serious limitation, which should be mentioned in discussion. It is good that You have provided METs from Your study for comparison. Could You describe method of measurements of METs used in the paper?

And from the last review, still unresolved:

“bor-323 derline significant result” even if Authors themselves use such description, You should not use it. If Authors assumed alpha level on 0.05, it means that the result is not statistically significant.

Author Response

Major issues:

  1. „Previous research suggests aerobic exercise-induced increases in skeletal muscle PGC-1α1 may be an important regulator of KAT expression in skeletal muscle, which, via modulation in plasma kynurenine levels, may alleviate stress-induced depression and promote hippocampal neuronal plasticity(Ruegsegger et al., 2018) [19]. In addition, facial muscles exercise activates the motor component of the ventral vagal complex, which originates in the nucleus ambiguus, regulates and coordinates the muscles of the face and head with the bronchi and heart. These connections help orient the person towards human connection and engagement in prosocial interactions and provide more flexible and adaptive responses to environmental challenges including social interactions (Sullivan et al., 2018 76 [20]).”

Please be more precise here. Is Your hypothesis is that some of the facial muscle exercise effects on some biochemical pathways are similar as in more traditional approach to aerobic training engaging large muscle groups as walking/running on treadmill cycling etc? What about the difference of amount of muscle groups engaged? Are the more complex movements more effective in most of the downstream biochemical effects? Is facial muscle exercise similar to continuous form of aerobic exercise or more to the interval training? What about those effects on autonomic nervous system? Could You be more specific? I suppose that the general consensus in the physiology of exercise is that during first seconds of aerobic exercise there is inhibition of vagus heart input, therefore heart rate could increase. With the further increase of exercise intensity there is increasingly stronger activation if sympathetic nervous system. So how facial muscle exercise activate ventral vagal complex? Could You explain that? Are You describing acute exercise effects session or long-term effects of exercise programme? Is the heart rate lowers during facial muscle exercise? So why there is an increase in METS as You state?
I suppose that facial muscle exercise might be a really interesting approach. Bot if You are referring to polyvagal theory as the mechanism underlying its effects, You should describe it more. It would be the best if You would be able to do so in comparison to more common forms of physical exercise. What about groups of patients that are not able to undergo “traditional” aerobic exercise training? Patients with CFS with sever post-exertional malaise, patients with movement disorders? Maybe they can be engaged is such training programme involving face muscles only?

Response

We thank the reviewer for reading our paper and providing the constructive feedback. In line with your valuable comment and question, we have added the information  in the Introduction section(highlighted in yellow).

  1. „Two investigators (RO and TM) independently searched for eligible studies in Pub-91 Med, Ichu-shi Web, which is a Japanese journal database[24] of Systematic Reviews”

It is nice that You have used more than one database. However, one can have an impression that You have not clearly phrased the aim of the search. Why did You have included database for systematic reviews? Were You looking for studies intervention bases studies? It would be the best to do systematic review on RCTs, however I suppose that there is lack of high quality studies in this area, what should be mentioned in the limitations. But why searching for systematic reviews? Is the current paper an umbrella review? Maybe it would be better for before-after intervention based studies? Or intervention based studies with control groups? What was the aim of the search? If not looking for systematic reviews, that why using database for systematic reviews?

Response

We apologize to mistranslated it. We don't have included database for systematic reviews. So we cut the part of the manuscript.

There are two table 1 in the current version of paper. Why put “Table 1. Classification standard of evidence level”?

Response

We apologize to mistranslated it. Another reviewer commented to provide evidence level criteria.

  1. Considering limitations of the analysed trials, Your conclusions “The outcomes of this systematic review suggest that voluntary facial muscle exercises may help to improve mental health” are too strong. What mental health indicators could be improved? Are those conclusions specific to psychiatric disorders? You should be more specific here, conclusions should be firmly based on You analysis. Please be as specific as You can.

Response

We thank the reviewer. In line with your valuable comment and question, we have modified the information in the “3.2 Interventions” section(highlighted in yellow).

Minor issues:

“reported that the intencity of facial muscles exercise is 3-3.5 METS( similar to brisk walk-157 ing) [31].”

Response
We apologize to mistranslated in. The intencity of our intervention program was 2.5-3METs. We have modified the intensity in the “4.2 Limitations of the review” section(highlighted in yellow).

Once again, this section needs to be improved. The idea is to show the reader the high variability in training programs specifics included in studied that You have analysed. For instance, there were some that lasted for two weeks. In literature on hypertrophic training 2 weeks would be too ow amount of time to show any significant improvement in muscle mass. On the other hand some training programme(s) lasted for 3 months. Maybe the best would be to calculate total amount of time (in minutes or hours) spent on training programme in each study? So if it was 2 weeks and 2 session for 60 minutes per week it would be 4 session in total for 60 minutes, what would give 4*60=240 minutes of total amount of time spend on exercise. Could You report that measure for each analysed study? Are You noting some relationship between time spend on training programmes and effects? Or maybe even 2 weeks could be effective in improving primary outcomes? What about the intensity of exercise? Is that was reported? If not, it is a serious limitation, which should be mentioned in discussion. It is good that You have provided METs from Your study for comparison. Could You describe method of measurements of METs used in the paper?

Response
We thank the reviewer. In line with your valuable comment and question, we have added the information in the “3.2 Interventions” section(highlighted in yellow).

And from the last review, still unresolved:

“borderline significant result” even if Authors themselves use such description, You should not use it. If Authors assumed alpha level on 0.05, it means that the result is not statistically significant.

Response
We thank the reviewer. We apologize that we skipped the previous review. In line with your definite comment, we have modified the information in the “3.6.5  Immunological and physiological effects” section(highlighted in yellow).

Reviewer 2 Report

I would like to thank you for the opportunity to review the paper entitled: Effects of Facial Muscles Exercise for Mental Health: A systematic Review. The article is very interesting and only minor changes have been identified, which I will now list: 1. Line 137, change the number 4 for letter. 2. Table 1 must not be an image. 3. There are two tables 1. 4. The font of the tables must be the same as in the text. It is interesting to make a table comparing the programs developed in the different papers. 5. It is necessary to apply Consensus on Exercise Reporting Template (CERT) (Slade, 2015)

Author Response

Comments of Reviewer2

I would like to thank you for the opportunity to review the paper entitled: Effects of Facial Muscles Exercise for Mental Health: A systematic Review. The article is very interesting and only minor changes have been identified, which I will now list: 1. Line 137, change the number 4 for letter. 2. Table 1 must not be an image. 3. There are two tables 1. 4. The font of the tables must be the same as in the text. It is interesting to make a table comparing the programs developed in the different papers. 5. It is necessary to apply Consensus on Exercise Reporting Template (CERT) (Slade, 2015)

Response

We thank the reviewer for reading our paper and providing the constructive feedback. In line with your valuable comment and question, we have added and modified the information(highlighted in yellow). But about list 5 is not modified. Because this systematic review was conducted according to the Preferred Reporting Items for Systematic Reviews and Meta-Analysis (PRISMA) statement[23].

Round 2

Reviewer 1 Report

Some minor issues were improved. However, in my opinion the are still some major issues. The most important seems to be drawing wrong/inappropriate conclusions from the analysed literature. It makes the current paper to be not scientifically valid in the current form.

From previous review:

Major issues:

1.„ In addition, facial muscles 71 exercise activates the motor component of the ventral vagal complex, which originates in 72 the nucleus ambiguus, regulates and coordinates the muscles of the face and head with 73 the bronchi and heart.

What about those effects on autonomic nervous system? Could You be more specific? I suppose that the general consensus in the physiology of exercise is that during first seconds of aerobic exercise there is inhibition of vagus heart input, therefore heart rate could increase. With the further increase of exercise intensity there is increasingly stronger activation if sympathetic nervous system. So how facial muscle exercise activate ventral vagal complex? Could You explain that? Are You describing acute exercise effects session or long-term effects of exercise programme? Is the heart rate lowers during facial muscle exercise? So why there is an increase in METS as You state?

2.„”3.1. Study selection and characteristics”

I do still see no changes in this subparagraph. What kind of databases were used? If PubMed only, then it should be mentioned in the limitations of the study in the discussion.

  1. What is the most important issue which cause that, in my opinion, the study could not be published yet in the present form: Considering limitations of the analysed trials, Your conclusions “The outcomes of this systematic review suggest that voluntary 269 facial muscle exercises may help to improve mental health” are too strong. What mental health indicators could be improved? Are those conclusions specific to psychiatric disorders? So for example does bipolar or schizophrenia could be treated? You should state more specific conclusion. I do strongly suggest to revise Table 1 (You do still have two “table 1” in Your manuscript). For instance, see study Chuch
    uen et al. 2015. There is one outcome measure „1)the Suanprung stress test-60 SPST-6”, but in column with results there are two results. My strong suggestion is to revise this table:
  2. To add explanation of every acronym under the table (what BDI means etc)
  3. To write down not about results but about conclusions from each study in a specific column (for example if BDI is a depression scale, and there was no significant improvement in BDI, that means that there were no significant improvement in depression severity etc.) Then, please describe conclusion of Your study based on the column with conclusions from particular studies. In fact, from analysed literature facial exercises were not tested on “all mental health problems” so it cannot be Your conclusion. You can write down in conclusions of Your study that influence of facial muscle training have been tested on (depression, self-esteem, etc). And in the second sentence please write down on which indicators there were a significant impact, and on which indicators statistically non-significant changes were noted.